# Evidences of histologic thrombotic microangiopathy and the impact in renal outcomes of patients with IgA nephropathy

**Precil Diego Miranda de Menezes Neves**[1☯]*, **Rafael A. Souza**[1☯], **Fábio M. Torres**[1], **Fábio A. Reis**[1], **Rafaela B. Pinheiro**[2], **Cristiane B. Dias**[1], **Luis Yu**[1], **Viktoria Woronik**[1], **Luzia S. Furukawa**[1], **Lívia B. Cavalcante**[2], **Stanley de Almeida Araújo**[3,4], **David Campos Wanderley**[3,4], **Denise M. Malheiros**[2], **Lectícia B. Jorge**[1]

1 Nephrology Division, University of São Paulo, School of Medicine, São Paulo, SP, Brazil, 2 Pathology Division, University of São Paulo, School of Medicine, São Paulo, SP, Brazil, 3 Nephropathology Institute, Belo Horizonte, MG, Brazil, 4 Pathology Division, Federal University of Minas Gerais, Belo Horizonte, MG, Brazil

☯ These authors contributed equally to this work.
* precilmed61@yahoo.com.br

## Abstract

### Introduction

IgA nephropathy (IgAN) is the most common primary glomerulopathy worldwide. According to the Oxford Classification, changes in the kidney vascular compartment are not related with worse outcomes. This paper aims to assess the impact of thrombotic microangiopathy (TMA) in the outcomes of Brazilian patients with IgAN.

### Materials and methods

Analysis of clinical data and kidney biopsy findings from patients with IgAN to assess the impact of TMA on renal outcomes.

### Results

The majority of the 118 patients included were females (54.3%); mean age of 33 years (25;43); hypertension and hematuria were observed in 67.8% and 89.8%, respectively. Median creatinine: 1.45mg/dL; eGFR: 48.8ml/min/1.73m$^2$; 24-hour proteinuria: 2.01g; low serum C3: 12.5%. Regarding to Oxford Classification: M1: 76.3%; E1: 35.6%; S1: 70.3%; T1/T2: 38.3%; C1/C2: 28.8%. Average follow-up: 65 months. Histologic evidence of TMA were detected in 21 (17.8%) patients and those ones presented more frequently hypertension (100% *vs*. 61%, *p* <0.0001), hematuria (100% *vs* 87.6%, *p* = 0.0001), worse creatinine levels (3.8 *vs*. 1.38 mg/dL, *p* = 0.0001), eGFR (18 *vs*. 60 ml/min/1.73m$^2$), *p* = 0.0001), low serum C3 (28.5% *vs*. 10.4%, *p* = 0.003), lower hemoglobin levels (10.6 *vs*. 12.7g/dL, *p*<0.001) and platelet counts (207,000 *vs*. 267,000, p = 0.001). Biopsy findings of individuals with TMA revealed only greater proportions of E1 (68% *vs*. 32%, *p* = 0.002). Individuals with TMA were followed for less time (7 *vs*. 65 months, p<0.0001) since they progressed more frequently to chronic kidney disease (CKD) requiring kidney replacement therapy (KRT) (71.4% *vs*. 21,6%, p<0.0001). Male sex, T1/T2, and TMA were independently associated with progression to CKD-KRT.

**Data Availability Statement:** All relevant data are within the manuscript.

**Funding:** The authors received no specific funding for this work.

**Competing interests:** The authors have declared that no competing interests exist.

## Conclusions

In this study patients with TMA had worse clinical manifestations and outcomes. In terms of histologic evidence, E1 distinguished patients with TMA from other patients. Further studies are necessary to analyze the impact of vascular lesions on IgAN prognosis.

## Introduction

IgA nephropathy (IgAN) is a highly prevalent condition worldwide and ranks as the most common primary glomerulopathy in some countries [1–3]. Given the high prevalence of the disease and the fact that about 30% of the patients with IgAN progress to chronic kidney disease (CKD) requiring kidney replacement therapy (KRT) [1,4], it is imperative to identify clinical and histology markers associated with worse renal outcomes.

The most widely accepted explanation for the IgAN pathogenesis is the 4-hit hypothesis, in which Hit 1 involves the production of hypoglycosylated IgA1; Hit 2 starts with the production of IgG antibodies that recognize hypoglycosylated IgA1; Hit 3 regards the formation of potentially nephritogenic IgG/IgA1 immune complexes; and in the Hit 4 there is deposition of formed complexes in the glomerular mesangium and capillaries, thereby activating the immune system and leading to the recruitment of inflammatory cells, cytokines, and the activation of the complement system [1,2,5].

The Oxford Classification (OC) [6,7] was first published in 2009 as an attempt to identify kidney biopsy alterations possibly associated with worse outcomes in patients with IgAN. Mesangial hypercellularity, segmental glomerulosclerosis, and interstitial fibrosis/tubular atrophy have been associated with progression to CKD-KRT, while endocapillary hypercellularity was first correlated with function decline in patients on immunosuppressant therapy and later with worse renal outcomes. An updated version of the Oxford Classification was published in 2017 [8], and cellular crescents were added as markers of worse renal outcomes. It should be mentioned that vascular alterations were not included in the Oxford Classification, since they were not associated with worse outcomes in patients with IgAN. However, recent studies [9–11] have looked into the role of vascular alterations and their ties with the outcomes of patients with IgAN, shedding light on a matter yet unresolved in the literature.

Thrombotic microangiopathy (TMA) is a histology finding of vascular involvement associated with some renal conditions–atypical and typical hemolytic-uremic syndrome, eclampsia, accelerated hypertension, thrombotic thrombocytopenic purpura–that may also be induced by certain drugs [12,13]. Histology and serum findings of TMA have been associated with other primary and secondary glomerulopathies–lupus nephritis, ANCA-associated vasculitis, focal segmental glomerulosclerosis, and IgA nephropathy–and correlated with worse renal outcomes in individuals with IgAN [9,10,14,15].

This study aimed to assess the impact of histologic findings of TMA on the renal outcomes of individuals with IgAN seen at a healthcare center in Brazil.

## Materials and methods

### Study design and population

This retrospective single-center study included patients diagnosed with IgA nephropathy based on kidney biopsy findings from 2000 to 2018. Patients with IgAN secondary to systemic conditions (Henoch-Schönlein purpura, liver disease, autoimmune disease, HIV infection)

and individuals with insufficient follow-up or outcome data were excluded, along with patients with fewer than eight glomeruli for analysis via the Oxford Classification.

Among our patients there are no solid organs transplanted patients, bone marrow transplant, pregnant women or patients diagnosed with cancer. We have no description of possible medications associated with TMA. Serologies were performed for Hepatitis B, C and HIV as well as research for autoimmune diseases (ANA, ANCA, Rheumatoid Factor, lupus anticoagulant). No genetic tests have been performed.

The following clinical data were considered at the time of kidney biopsy: age; sex; serum creatinine (SCr); estimated glomerular filtration rate (e-GFR); 24-hour proteinuria and/or urine protein/creatinine (UPC) ratio; hematuria; hypertension; serum C3 level; serum IgA; hemoglobin; platelet count; lactate dehydrogenase (LDH); and indirect bilirubin. The glomerular filtration rate was estimated based on the CKD-EPI [16] equation. Hematuria was defined as ≥3 red blood cells/high-power field in a sample of urine. Hypertension was defined as a blood pressure ≥140 and/or 90mmHg [17]. The reference ranges laboratory tests were as follows: C3 (90-180mg/dL); IgA (69-382mg/dL); LDH (135-214U/L); indirect bilirubin (0.2; 0.8mg/dL); haptoglobin (30-200mg/dL). Anemia was defined as hemoglobin <12g/dL for females and 13g/dL for males [18]. Thrombocytopenia was defined as having a platelet count <150,000/mm$^3$ [19]. Presence of schistocytes was verified via peripheral blood smear tests.

Patients were also analyzed for prescription of angiotensin-converting-enzyme (ACE) inhibitors or angiotensin II receptor blockers (ARBs), corticosteroids, and other immunosuppressants. The end of follow-up was defined either by the last visit of the patient to the healthcare unit or by referral to dialysis or kidney transplantation.

## Histopathology

Kidney biopsy specimens analyzed by light microscopy were stained by hematoxylin and eosin (H&E), Masson's trichrome, periodic acid–Schiff (PAS), and periodic acid silver methenamine stain (PAMS). Analysis by immunofluorescence microscopy included serum anti-IgA, IgG, IgM, C3, C1q, kappa and lambda light chains, and fibrinogen, with positive deposition defined when intensity ≥1. The biopsies were reviewed by a renal pathologist and classified based on the latest version of the Oxford Classification [8].

As previously described in the literature [20], acute TMA histologic findings were categorized based on renal compartments. In glomeruli: presence of thrombi, edema, or endothelial denudation, fragmented red blood cells, mesangiolysis, microaneurysms; in arterioles: thrombi, edema, or endothelial denudation, intramural fibrin, fragmented red blood cells, edema of the intima, myocyte necrosis; and in arteries: thrombi, myxoid intimal change, intramural fibrin, fragmented red blood cells. Chronic findings included in glomeruli: double contour in capillaries with mesangial interposition; in arterioles hyaline deposits; in arteries fibrous intimal thickening and concentric lamination resembling the bulb of an onion. Histologic findings of TMA were assessed only based on light microscopy examination.

## CD68 and C4d immunohistochemistry

Formalin-fixed, paraffin-embedded tissue was sectioned at 2μm and stained with rabbit monoclonal anti-CD68 (Santa Cruz Biotechnology) and anti-C4d (Clone SP91, Spring-Bioscience) as the primary antibody. All cases were stained by hand using routine protocols, including deparaffinization, followed by antigen retrieval (tissue section was boiled in 1mM EDTA, pH 8.0 for 10 min followed by cooling at room temperature for 20 min), protein blocking (DPB-125S; Spring, Pleasanton, CA, USA), incubation for primary antibody at room temperature for 30 min (1:40) and for secondary goat anti-rabbit IgG (DHRR-999; Spring, Pleasanton, CA,

USA) at 1:360. Detection was performed with streptavidin/horseradish peroxidase (SPB-125; Spring) and developed with Stable DAB (Spring). CD68-positive cells in the glomeruli and tubulointerstitium were quantified and the final count was expressed as number of cells/glomerulus and cells/field, respectively. C4d immunohistochemistry was considered positive if glomerular staining was observed.

### Endpoint analysis

Progression to CKD-KRT was the primary endpoint assessed. Secondary endpoint included achieving an e-GFR $\leq$60ml/min/1.73m$^2$.

### Statistical analysis

The distribution of variables was assessed with the Shapiro-Wilk test. Qualitative variables were expressed as proportions and compared against each other via the chi-squared test or Fisher's exact test. Variables following a parametric distribution were expressed as mean values ± standard error and compared against each other with Student's t-test. Variables with non-parametric distributions were expressed as median values (first and third quartiles) and compared against each other with the Mann-Whitney U test. Logistic multinomial regression was performed in all patients and it was used in multivariate analysis to assess independent risk factors to and adjusted for sex, hypertension, creatinine >1.2mg/dL (upper normality range of the test), Oxford classification parameters (M, E, S, T, C) and TMA. Statistical significance was attributed to significance level alpha = 0.05. Survival free of dialysis in patients with and without evidence of TMA in kidney biopsy specimens was analyzed by Kaplan-Meier curve.

## Results

Exclusion criteria accounted for the removal of 50 patients from the original population of 168 individuals. Of the remaining 118 patients, 65 were females (55%), 86 were whites (73%), 25 were black (21%), 7 were east-asian (6%); 80 (67.8%) presented with hypertension and 106 (89.8%) hematuria (89.8%). Low serum C3 was detected in 15 (12.5%) patients. Table 1 shows patient data and laboratory tests at the time of kidney biopsy. Histologic findings based on the Oxford Classification were as follows: M1 (79.6%), E1 (35.6%), S1 (70.3%), T1/T2 (38.3%), and C1/C2 (28.8%). The median follow-up time was 65 months, and 36 individuals progressed to CKD-KRT.

Histologic evidences of TMA were seen in 21 (17.8%) patients and acute findings as well as chronic lesions predominate in arterioles over glomeruli. More frequent acute lesions in arteriolar compartment were intramural fibrin (33.3%) and myocyte necrosis (23.3%) while fibrous intimal thickening with concentric lamination hyaline deposits (100%) and hyaline deposits (90.5%) were the most common chronic lesions. Glomerular compartment showed more chronic lesions, such as double contour in capillaries with mesangial interposition in 38.1% of the patients, than acute findings such as edema or endothelial denudation (14.3%) and mesangiolysis (14.3%). The Table 2 illustrates histological findings of TMA on renal biopsy. Table 3 describes the presence of serum findings consistent with TMA in patients with and without histologic evidence of TMA. When compared to individuals without signs of TMA in kidney biopsy, patients with histologic evidence of TMA had more anemia (81.3% *vs*. 24.7%, p = 0.001), more schistocytes in peripheral blood (44.5% *vs*. 3.1%, p = 0.0007), and were more prone to developing thrombocytopenia, albeit not significantly (14.3% *vs*. 4.1%, p = 0.07).

The analysis of patients with and without TMA on kidney biopsy (Table 4) revealed that the first group had no difference in prevalence or progression to ESKD regarding the race, had a

**Table 1. Baseline clinical characteristics and kidney biopsy findings of patients with IgA Nephropathy.**

| | N = 118 |
|---|---|
| Age (years) | 33 (25;43) |
| Female sex (n/%) | 65 / 55 |
| Race (n/%) | |
| White | 86 / 73 |
| Black | 25 / 21 |
| East-asian | 7 / 6 |
| Serum creatinine (mg/dL) | 1.45 (0.99;2.6) |
| e-GFR by CKD-EPI (ml/min/1.73m$^2$)[b] | 48.8 (27.5;78.5) |
| 24h-proteinuria (g) | 2.01 (1.1;3.7) |
| Serum albumin (g/dl) | 3.4 (2.9;3.8) |
| Hematuria (n/%) | 106 / 89.8 |
| Hypertension (n/%) | 80 / 67.8 |
| Low serum C3 levels (n/%) | 15 / 12.5 |
| Follow-up (months) | 65 (27;115) |
| ΔeGFR (ml/min/1.73m$^2$/year) | -1.25 (-7.11;0.91) |
| CKD-KRT (n/%) | 36 / 30.5 |
| Time to CKD-KRT (months) | 9 (3;38) |
| Kidney Histology—Oxford Classification (n/%) | |
| M1 | 94 / 79.6 |
| E1 | 42 / 35.6 |
| S1 | 83 / 70.3 |
| T1/T2 | 45 / 38.3 |
| C1/C2 | 34 / 28.8 |
| Thrombotic Microangiopathy (n/%) | 21 / 17.8 |

*eGFR*: Estimated glomerular filtration rate, *CKD requiring RRT*: Chronic Kidney Disease requiring Renal Replacement Therapy, *M1*: Mesangial hypercellularity, *E1*: Endocapillary hypercellularity, *S1*: Segmental glomerulosclerosis, *T1/T2*: Tubular atrophy or interstitial fibrosis, *C1/C2*: Cellular crescent.

greater proportion of hypertensive individuals (100 *vs.* 61, p<0.0001) and with hematuria (100 *vs.* 87.6, p = 0.0001). Patients with TMA had worse median serum creatinine levels (3.8 *vs.* 1.38 mg/dL, p = 0.0001), eGFR (18 *vs.* 60.2 ml/min/1.73m$^2$, p = 0.0001), and more frequent low serum C3 (28.5 *vs.* 10.4, p = 0.003). There was no difference in the treatment prescribed to the two groups of patients. In regard to the histology parameters of the Oxford Classification, solely a higher proportion of individuals with E1 was observed (68% vs. 32%, p<0.002). Patients with histologic evidence TMA were followed for less time (7 *vs.* 65 months, p<0.0001), since a greater portion of them progressed to CKD-KRT (71.4% *vs.* 21.6%, p<0.0001) and in less time (3 *vs.* 16 months, p = 0.003) on account of the quicker eGFR decrease they experienced (-6.8 *vs.* -0.65 ml/min/1.73m$^2$/year, p = 0.01).

The comparison of patients that progressed or not to CKD-KRT (Table 5) revealed that most of the individuals in the first group were males (63.8% *vs.* 36.5%, p = 0.01), younger (30 *vs.* 34 years of age, p = 0.04), had hypertension (86.1% *vs.* 56.1%, p = 0.0016), worse creatinine levels at the biopsy time (3 *vs.* 1.2mg/dL, p<0.0001), lower eGFR (22.5 *vs.* 64.8ml/min/1.73m$^2$, p<0.0001), and more frequent low serum C3 (25.7% *vs* 7,3%, p = 0.01), without difference in proteinuria. Kidney biopsy findings pointed to a greater proportion of patients with T1/T2 (54.2% *vs.* 23,2%, p<0.0002) and a greater proportion of individuals with TMA (41.7% *vs.* 7,32%, p<0.0001). No difference was seen in the treatment prescribed to both patient groups.

**Table 2. Histological findings of thrombotic mycroangiopathy in renal biopsies of IgAN patients.**

|  | N = 21 |
|---|---|
| **Acute Findings (n/%)** |  |
| Glomeruli |  |
| Glomeruli thrombi | 2 / 9.5 |
| Edema or endothelial denudation | 3 / 14.3 |
| Fragmented red blood cells | 1 / 4.8 |
| Mesangiolysis | 3 / 14.3 |
| Microaneurysms | 0 / 0 |
| Arterioles |  |
| Thrombi | 3 / 14.3 |
| Edema or endothelial denudation | 4 / 19 |
| Intramural fibrin | 7 / 33.3 |
| Fragmented red blood cells | 1 / 4.8 |
| Edema of the intima | 0 / 0 |
| Myocyte necrosis | 5 / 23.8 |
| Arteries |  |
| Thrombi | 0 / 0 |
| Myxoid intimal swelling | 3 / 14.3 |
| Intramural fibrin | 0 / 0 |
| Fragmented red blood cells | 0 / 0 |
| **Chronic findings (n/%)** |  |
| Glomeruli |  |
| Double contour in capillaries with mesangial interposition | 8 / 38.1 |
| Arterioles |  |
| Hyaline deposits | 19 / 90.5 |
| Arteries |  |
| Fibrous intimal thickening with concentric lamination | 21 / 100 |

Patients progressing to CKD-KRT were followed for less time (7 *vs*. 69 months, p<0.0001) on account of more pronounced eGFR decreases (-8.17 *vs*. -0.21 ml/min/1.73m$^2$/ano, p<0.0001).

Immunohistochemistry for CD68 (Fig 1) was performed to look into macrophage-mediated tissue inflammation in 76 patients, nine from the group of 21 patients with TMA (43%) and 67 without TMA (69.1%). Fig 1 shows immunohistochemistry images derived from renal biopsy specimens. No statistically significant difference was found when patients with and

**Table 3. Laboratory findings of thrombotic microangiopathy in patients with and without histologic evidence of TMA.**

|  | TMA (n = 21) | No-TMA (n = 97) | *p* |
|---|---|---|---|
| Anemia* (n/%) | 13 / 61.9 | 24 / 24.7 | 0.002 |
| Low Platelet count (n/%) | 3 / 14.3 | 4 / 4.1 | 0.07 |
| High serum LDH level (n/%) | 13 / 61.9 | 45 / 46.4 | 0.29 |
| Low serum haptoglobin level (n/%) | 2 / 9.5 | 2 / 2.6 | 0.29 |
| High serum indirect bilirubin level (n/%) | 0 / 0 | 4 / 4.12 | 0.77 |
| Schistocytes on peripheral blood smear (n/%) | 4 / 19 | 3 / 3.1 | 0.02 |

LDH: Lactate Dehydrogenase.

* Defined as hemoglobin <12g/l for females and <13g/l for males [20].

**Table 4. Analysis of clinical parameters and laboratory findings in IgAN patients with and without histologic evidence of TMA.**

| | TMA (n = 21) | No-TMA (n = 97) | p |
|---|---|---|---|
| Male (n/%) | 10 / 47.6 | 43 / 44.3 | 0.62 |
| Age (years) | 32 (27;41) | 33 (24;44) | 0.83 |
| Race (n/%) | | | 0.653 |
| White | 14 / 16.3 | 72 / 83.7 | |
| Black | 6 / 24 | 19 / 76 | |
| East-asian | 1 / 14.3 | 6 / 85.7 | |
| Hypertension (n/%) | 21 / 100 | 59 / 61 | <0.0001 |
| Hematuria (n/%) | 21 / 100 | 85 / 87.6 | 0.0001 |
| Serum creatinine (mg/dL) | 3.8 (2.2;5.8) | 1.38 (0.91;1.9) | 0.0001 |
| e-GFR by CKD-EPI (ml/min/1,73m$^2$)[a] | 18.3 (9.2;30.5) | 60.2 (35.1;87.5) | 0.0001 |
| 24h-proteinuria (g) | 1.9 (0.9;3.96) | 2 (1.3;3.6) | 0.86 |
| Serum albumin (g/dL) | 3.2 (2.55;3.9) | 3.5 (3.1;3.8) | 0.26 |
| Low serum C3 levels (n/%) | 6 / 28.5 | 9 / 10.4 | 0.003 |
| Treatment | | | |
| ACE inhibitor or ARB (n/%) | 19 / 89.4 | 78 / 80.4 | 0.37 |
| Corticosteroids (n/%) | 11 / 52.6 | 63 / 64.7 | 0.35 |
| Other immunosuppressants (n/%) | 8 / 36.8 | 34 / 35.2 | 0.9 |
| Kidney Histology–Oxford Classification (n/%) | | | |
| M1 | 19 / 89.7 | 75 / 77.3 | 0.23 |
| E1 | 14 / 68 | 31 / 32 | 0.002 |
| S1 | 16 / 78.9 | 70 / 72.1 | 0.54 |
| T1/T2 | 12 / 57.9 | 35 / 36.1 | 0.07 |
| C1/C2 | 8 / 32.1 | 26 / 26.8 | 0.57 |
| Immunofluorescence positivity (n/%) | | | |
| IgM | 6 / 28.5 | 23 / 23.7 | 0.84 |
| IgG | 0 / 0 | 10 / 10.3 | 0.26 |
| C3 | 18 / 85.7 | 75 / 77.3 | 0.57 |
| C1q | 0 / 0 | 7 / 7.2 | 0.44 |
| CD68 Immunohistochemistry (cells/field) | | | |
| Glomeruli | 4.3 (3.02;6.0) | 2.25 (1.56;5) | 0.12 |
| Tubulointerstitium | 25.4 (14;34.7) | 18.3 (9.8;28.3) | 0.25 |
| C4d glomerular staining (n/%)[c] | 7 / 77.8 | 26 / 41.3 | 0.04 |
| Follow-up (months) | 7 (3;21) | 65 (27;115) | <0.0001 |
| ΔeGFR (ml/min/1,73m$^2$/year) | -6.8 (-24;0) | -0.65 (-4.48;2.3) | 0.01 |
| CKD-KRT (n/%) | 15 / 71.4 | 21 / 21.6 | <0.0001 |
| Time to CKD-KRT (months) | 3 (3;7) | 16 (4;64) | 0.003 |

*e-GFR* estimated glomerular filtration rate, *M1*: Mesangial hypercellularity, *E1*: Endocapillary hypercellularity, *S1*: Segmental glomerulosclerosis, *T1/T2*: Tubular atrophy or interstitial fibrosis, *C1/C2*: Cellular crescent, *CKD requiring RRT*: Chronic Kidney Disease requiring Renal Replacement Therapy.

[a]As determined by the Chronic Kidney Disease–Epidemiology Collaboration equation.

[c] Analysis of immunohistochemistry for C4d was possible in 72 patients, 9 from the group with TMA (43%) and 63 without TMA (64.9%).

without TMA were compared for number of CD68 positive glomerular (4.3 *vs*. 2.25, p = 0.12) or interstitial (25.4 *vs*. 18.3, p = 0.25) cells. The comparison between patients progressing or not to CKD-KRT did not yield significant differences for glomerular cells labeled positive for CD68 (3.35 *vs*. 2.24, p = 0.38), but significantly more CD68 interstitial cells were seen in individuals progressing to CKD-KRT (32.5 *vs*. 15.6, p<0.0001) (Table 4). In total, 72 kidney

**Table 5. Clinical parameters and laboratory findings of IgAN patients progressing or not to CKD-KRT.**

| | CKD-KRT (n = 36) | No CKD-KRT (n = 82) | p |
|---|---|---|---|
| Age (years) | 30 (24;40) | 34 (26;50) | 0.04 |
| Male sex (n/%) | 23 (63,8) | 30 (36,5) | 0.01 |
| Creatinine (mg/dL) | 3 (2.3;5.6) | 1.2 (0.9;1.7) | <0.0001 |
| eGFR (ml/min/1.73m$^2$) | 22.5 (9.6;36) | 64.8 (40;91.7) | <0.0001 |
| Proteinuria (g/day) | 2.4 (1.3;4.1) | 1.58 (1;2.98) | 0.07 |
| Albumin (g/dL) | 3.4 (2.7;3.8) | 3.5 (3;3.9) | 0.61 |
| Hematuria (n/%) | 32 / 88.9 | 70 / 85.3 | 0.77 |
| Hypertension (n/%) | 31 / 86.1 | 46 / 56.1 | 0.0016 |
| Consumption of C3 (n/%) | 9 / 25.7 | 6 / 7.3 | 0.01 |
| Follow-up (months) | 7 (3;39) | 69 (35;122) | <0.0001 |
| ΔGFR (ml/min/1.73m$^2$/year) | -8.17 (-31;2.46) | -0.21 (-2.1;2.7) | <0.0001 |
| Kidney Histology–Oxford Classification (n/%) | | | |
| M1 | 21 / 58.33 | 60 / 73.1 | 0.13 |
| E1 | 16 / 44.4 | 24 / 29.2 | 0.13 |
| S1 | 24 / 66.6 | 51 / 62.2 | 0.68 |
| T1/T2 | 20 / 54.2 | 19 / 23.2 | 0.002 |
| C1/C2 | 13 / 36.1 | 18 / 21.9 | 0.11 |
| Immunofluorescence positivity (n/%) | | | |
| IgM | 11 / 30.5 | 18 / 21.9 | 0.44 |
| IgG | 3 / 8.3 | 7 / 8.53 | 0.74 |
| C3 | 29 / 80.5 | 64 / 78 | 0.95 |
| C1q | 2 / 5.5 | 5 / 6.1 | 0.75 |
| CD68 Immunohistochemistry (cells/field) | | | |
| Glomeruli | 3.35 (1.87;5.82) | 2.24 (1.47;5.04) | 0.38 |
| Tubulointerstitium | 32.5 (24.6;51.9) | 15.6 (8.99;24.4) | <0.0001 |
| TMA (n/%) | 15 / 41.7 | 6 / 7.32 | <0.0001 |
| Treatment | | | |
| ACE inhibitor or ARB (n/%) | 17 / 47.2 | 54 / 65.8 | 0.06 |
| Corticosteroids (n/%) | 13 / 36.1 | 40 / 48.8 | 0.23 |
| Other immunosuppressants (n/%) | 8 / 22.2 | 16 / 19.6 | 0.81 |

*eGFR* estimated glomerular filtration rate. *SD* standard deviation. *M1*: Mesangial hypercellularity. *E1*: Endocapillary hypercellularity. *S1*: Segmental glomerulosclerosis. *T1/T2*: Tubular atrophy or interstitial fibrosis. *C1/C2 cellular* crescent. *ACEi* angiotensin-converting enzyme inhibitor. *ARB* angiotensin II receptor blocker *CKD-RRT*: Chronic Kidney Disease requiring Kidney Replacement Therapy.

[a]As determined by the Chronic Kidney Disease–Epidemiology Collaboration equation.

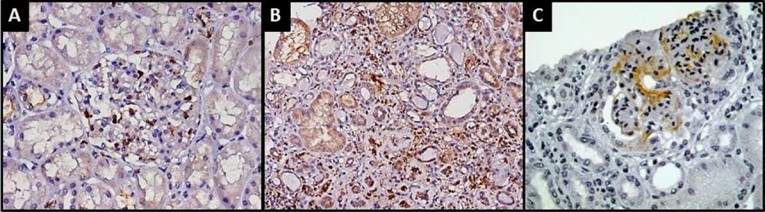

**Fig 1.** CD68 immunohistochemistry of kidney biopsies of IgAN patients labeled positive in the A) glomerulus (x400) and B) tubulointerstitium (x200). C) C4d immunohistochemistry revealing positive glomerular staining (x400).

**Table 6. Logistic regression analysis for the primary outcome[a] adjusted for sex, hypertension, creatinine, Oxford classification parameters (M,E,S,T,C) and TMA.**

| Variable | HR | 95% Confidence Interval for HR | | p |
|---|---|---|---|---|
| | | Lower | Upper | |
| Female | 0.24 | 0.07 | 0.54 | 0.03 |
| Hypertension | 1.02 | 0.19 | 5.36 | 0.98 |
| Creatinine >1.2mg/dl | 3.12 | 0.51 | 19.03 | 0.21 |
| M1 | 1.53 | 0.28 | 8.52 | 0.62 |
| E1 | 0.56 | 0.14 | 2.27 | 0.41 |
| S1 | 10.2 | 0.88 | 118.1 | 0.06 |
| T1/T2 | 8.17 | 2.17 | 30.89 | 0.002 |
| C1/C2 | 3.62 | 0.87 | 14.94 | 0.07 |
| TMA | 8.34 | 1.66 | 41.96 | 0.01 |

*HR* hazard ratio. e-GFR estimated glomerular filtration rate M1: Mesangial hypercellularity. E1: Endocapillary hypercellularity. S1: Segmental glomerulosclerosis. T1/T2: Tubular atrophy or interstitial fibrosis. C1/C2 cellular crescent. *CI* confidence interval. *T1/T2* mild to severe tubular atrophy or interstitial fibrosis. *TMA*: Thrombotic Microangiopathy.

[a]CKD requiring RRT: Chronic Kidney Disease requiring Renal Replacement Therapy.

biopsies were available to stain for C4d, including 9 tissue sections from patients with TMA and 63 tissue sections from those without TMA. Glomerular deposits of C4d were more frequent in patients with TMA as compared to those without TMA (n/M = 7/9 and n/N = 26/63; P = 0.04).

Logistic regression performed to assess risk factors independently related to the primary end point in all the 118 patients found that female sex was protective against the condition (Hazard Ratio—HR 0.24; 95% CI 0.07–0.54, *p* = 0.03), while T1/T2 scores (HR: 8.17; 95% CI 2.17–30.89, *p* = 0.001) and histologic signs of TMA (HR: 8.34; 95% CI 1.66; 41.96, p<0.01) were associated with progression to CKD-KRT (Table 6). Fig 2 shows the differences in renal survival for patients with and without evidences of TMA in kidney biopsy.

## Discussion

Some clinical findings associated with worse outcomes in individuals with IgAN–male sex, older age, persistent microscopic hematuria, hypertension, proteinuria, and creatinine levels on kidney biopsy [21–24]–have been described as a factor in the progression to CKD-KRT within five years seen in approximately 30% of the individuals with the condition [4,22,25]. In light of this fact and the variety of histologic findings to consider, the need to standardize the analysis of kidney biopsy of IgAN patients and the search for histology-related prognostic factors led to the creation of histologic classifications [6–8,26], with the Oxford Classification standing as the most widely accepted among pathologists and nephrologists. However, the OC does not consider TMA despite its role in disease progression and although it has been described to affect 2.2% to 53% of the individuals with IgAN [11,26]. Discrepancies in the frequency of involvement by TMA may be ascribed to differences among populations, although the adoption of diverse criteria to diagnose the condition–some including electron microscopy examination and immunohistochemistry staining protocols–is a relevant matter to consider.

El Karoui *et al* [11] studied a multi-center population of patients with IgAN and TMA. The author found that 68 (53.15%) of 128 patients had histologic evidence of TMA, a proportion higher than the reported in other studies. Diagnosis of TMA was based on light microscopy

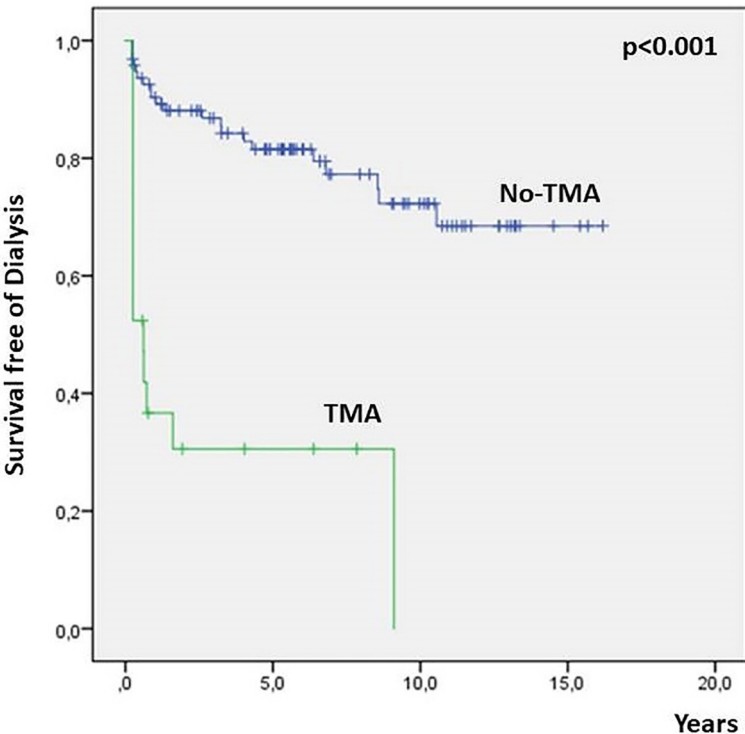

**Fig 2. Survival free of dialysis of patients with IgAN with and without evidence of TMA in kidney biopsy specimens.**

and immunohistochemistry staining for CD61. Patients with TMA progressed more frequently to CKD-KRT. A multicenter study carried out in Spain [27] included patients with IgAN and malignant hypertension. In the study, 13/186 patients (7%) were diagnosed with malignant hypertension. Patients with the condition were predominantly men in the fourth decade of life. Ten patients were on KRT at the end of the follow-up period. Three of the six individuals submitted to kidney transplantation returned to dialysis due to chronic allograft nephropathy associated with recurrent IgAN, although they did not have signs of malignant hypertension. Cai *et al* [10] studied 944 patients with IgAN from a single center in China, in which 194 patients with histologic evidence of TMA were found based on light and electron microscopy examination. Patients with TMA were older than their counterparts. Most were males with worse renal function, more proteinuria, and, interestingly enough, no differences in serum TMA markers. As seen in previous literature reports, patients with TMA reached the target endpoint more frequently and quickly. Haas *et al* [26] looked into 2290 patients with IgAN and found 49 (2.2%) with TMA based on light and electron microscopy examination. The patients in this group were younger and had worse renal function and proteinuria on admission. They were not analyzed for progression to CKD-KRT. Other studies support these findings [10,28–31]. Our study found histologic evidence of TMA in 17.8% of the included patients, confirming some of the clinical and workup findings described previously in the literature (more individuals with hypertension, worse renal function, and worse outcomes) [9–11]. As reported in other studies [9,10], patients often do not present with clinical or workup findings indicative of TMA. Therefore, in our study only anemia and increased schistocyte counts stood out in the comparison of individuals with and without histologic evidence of TMA, since the groups were not different in terms of platelet counts, serum LDH, or haptoglobin levels.

Among the etiologies associated with TMA we must consider malignant hypertension, thrombotic thrombocytopenic purpura, atypical hemolytic uremic syndrome (aHUS), shiga toxin-producing Escherichia coli (STEC-HUS), scleroderma renal crisis, hypertensive pregnancy disorders, antiphospholipide syndrome, infections, drug toxicity and metabolic diseases [12,32]. All of those conditions described above can manifest themselves as microangiopathic hemolytic anemia, with presentation in the form of hypertensive urgency/emergency and the differential diagnosis becomes important since the treatment varies according the base disease [33,34]. Timermans et al [35] analyzed a series of 9 patients with diagnostic hypertension-associated TMA and compatible findings at funduscopic examination. An interesting and high relevant finding of this study is that 67% of the patients had mutations in genes associated with aHUS, despite the neither absence of a family history of TMA or compatible hematological markers of the syndrome in most patients. Facing the high frequency of complement disorders in these cases, the authors highlight the importance to perform genetic tests in patients with malignant hypertension without evidences of secondary causes, since the progression to CKD-KRT is frequent and specific treatment can prevent recurrence of the disease after transplant. Such findings are supported by the studies of El Karoui et al [36] and Cavero et al [37] which showed that cases of TMA that present with hypertensive emergency are more often associated to complement disorders than other secondary etiologies.

Cases of IgAN associated with complement factor mutation have been described [31,38–41], some of which effectively treated with Eculizumab [32]. What is not known is whether patients with IgAN and TMA actually present with a combination of the two diseases or if immunoglobulin deposition associated with the onset of the disease might produce local manifestations of TMA. The early pathogenesis of TMA has been linked to endothelial lesion [41,42]. Studies have shown that in addition to tropism by immunoglobulin degradation and complement factors in the mesangium, patients with IgAN also produce anti-endothelial cell antibodies [43,44], thereby causing endothelial lesion and dissociation of the endothelium and the glomerular basement membrane, activation of the immune system (cytokines, inflammatory cells, complement system) and the coagulation cascade, producing local glomerular thrombosis [41,45,46]. On the other hand, the complement system may be activated via the lectin pathway as hypoglycosylated IgA1 –linked to the pathogenesis of IgAN–binds to endothelial cells [47–50]. This finding may be demonstrated by the overwhelming positivity for C4d in patients with IgAN and TMA without co-deposition of C1q [10,41]. Both theories are plausible, since it is not rare to find patients with only histologic and no clinical signs of TMA. We did not observe differences in the profile or intensity of immunoglobulin deposition (IgG, IgM) or complement proteins (C3, C1q) in our series between patients with and without TMA or individuals that progressed versus patients that did not progress to CKD-KRT. However, none of the patients with TMA had deposition of IgG or C1q, while 10.3% of the patients without TMA had deposition of IgG and 7.3% of C1q. Important contributions to IgAN pathological understanding of complement disturbances came from histochemical studies of kidney tissue C4d staining. On clinical settings C4d deposition has been associated with aggressive histology and worse clinical course [51,52]. Drachenberg et al [53] showed positive mesangial staining in 13 out of 34 IgAN patients and additional positivity in capillary walls in 9. Mesangial staining did not correlate with any of the Oxford scores while capillary wall staining correlated with E1 (endocapillary hypercellularity). Considering TMA patients, predominantly with aHUS diagnosis, a well-known complement disturbance condition, in the same study of Drachenberg et al [53], glomerular and/or arteriolar TMA lesions stained positive for C4d in all cases (100%) Chua et al [9] employed markers not used in routine clinical practice to assess the complement system and analyze depositions of C4d, C5b-9, mannose-binding lectin and Factor B in the renal tissue of patients with IgAN with and without TMA. The authors found

an association between C4d and TMA and the presence of the two conditions with worse renal survival, thus stressing the role of the complement system in the pathogenesis of TMA in patients with IgAN. Similar findings were observed in our patients, where the group with TMA expressed more frequently glomerular staining for C4d than those without TMA. Since only 2 of 10 patients with TMA had negative immunostaining for C4d, we opted for not analyze renal survival curve for the three groups (no-TMA vs TMA/C4d+ vs TMA/C4d-).

By approaching the complement system from a systemic standpoint, we found, in our series, more patients with low C3 levels in the TMA group when compared with individuals without TMA and with the group of patients that progressed to CKD requiring KRT. This finding is not reflected in renal tissue, in which C3 deposition did not occur differently in patients with or without TMA (85.7 *vs.* 77.3, p = 0.57) or in individuals progressing or not to CKD-KRT (80.5 *vs.* 78.0, p = 0.95). In regard to MEST-C and TMA parameters, we found associations only with parameter E, which identifies active glomerular endothelial lesions in association with findings consistent with TMA. Since parameter E is modifiable via treatment, it is important to notice that no significant difference was seen in the treatment prescribed to the two groups of patients.

Recent studies [2,49] pointed to the relevance of glomerular tissue expression of CD68 as an additional marker that allows pathologists to more accurately define parameter E for endocapillary hypercellularity, as described by Soares *et al* [49]. In our series, this marker did not elicit differences in the number of cells labeled for CD68 in the glomeruli or tubulointerstitium of patients with and without TMA. However, expression was increased in the tubulointerstitium of patients progressing to CKD requiring KRT on account of the histologic evidence of chronic disease described previously by Soares et al [49].

The limitations of this study derive from the fact that is was carried out in a single center and that diagnostic examination for TMA was based only on light microscopy, without the aid of immunohistochemistry staining or electron microscopy. However, studies performed in a single center are known for more consistent compliance with procedure, follow-up, and treatment processes. We are starting in our institution a study of genetic analysis of patients with TMA associated to primary and secondary glomerulopathies. At the moment, patients were not tested but we plan to perform analysis by whole exome sequencing and results will be performed as soon as those are available.

In conclusion, our study found that patients with IgAN and histologic evidences of TMA had clinically worse kidney function, more hypertension and hematuria, greater proportions of low serum C3 at kidney biopsy, with a larger amount of individuals with endocapillary hypercellularity (E1). Histologic evidence of TMA were not concurrent with clinical/laboratory markers of the condition. Patients with histologic evidence of TMA progressed more frequently and quickly to CKD-KRT when compared to individuals without TMA. Having signs of TMA in kidney biopsy specimens was an independent marker for progression to CKD-KRT when compared to other histologic predictors in the Oxford Classification. Additional studies are required to investigate the role of the complement system in TMA and IgAN and support the development of new therapeutic targets.

## Author Contributions

**Conceptualization:** Precil Diego Miranda de Menezes Neves, Viktoria Woronik, Luzia S. Furukawa, Lívia B. Cavalcante, Denise M. Malheiros, Lectícia B. Jorge.

**Data curation:** Precil Diego Miranda de Menezes Neves, Rafael A. Souza, Fábio M. Torres, Fábio A. Reis, Rafaela B. Pinheiro, Cristiane B. Dias, Luis Yu, Viktoria Woronik, Lívia B. Cavalcante, Lectícia B. Jorge.

**Formal analysis:** Precil Diego Miranda de Menezes Neves, Rafaela B. Pinheiro, Luis Yu, Viktoria Woronik, Luzia S. Furukawa, Lívia B. Cavalcante, Stanley de Almeida Araújo, David Campos Wanderley, Denise M. Malheiros, Lectícia B. Jorge.

**Investigation:** Precil Diego Miranda de Menezes Neves, Rafael A. Souza, Fábio M. Torres, Fábio A. Reis, Rafaela B. Pinheiro, Cristiane B. Dias, Luis Yu, Viktoria Woronik, Luzia S. Furukawa, Lívia B. Cavalcante, David Campos Wanderley, Denise M. Malheiros, Lectícia B. Jorge.

**Methodology:** Precil Diego Miranda de Menezes Neves, Rafael A. Souza, Fábio M. Torres, Fábio A. Reis, Rafaela B. Pinheiro, Cristiane B. Dias, Luis Yu, Viktoria Woronik, Luzia S. Furukawa, Lívia B. Cavalcante, Stanley de Almeida Araújo, David Campos Wanderley, Denise M. Malheiros, Lectícia B. Jorge.

**Supervision:** Cristiane B. Dias, Luis Yu, Viktoria Woronik, Stanley de Almeida Araújo, David Campos Wanderley, Denise M. Malheiros, Lectícia B. Jorge.

**Writing – original draft:** Precil Diego Miranda de Menezes Neves, Viktoria Woronik, Lectícia B. Jorge.

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
