## [Decision Letter · Decision Letter 0]

10 Jun 2020

PONE-D-20-12431

Evidences of histologic Thrombotic Microangiopathy and the impact in renal outcomes of patients with IgA nephropathy

PLOS ONE

Dear Dr. Neves,

Thank you for submitting your manuscript to PLOS ONE. After careful consideration, we feel that it has merit but does not fully meet PLOS ONE’s publication criteria as it currently stands. Therefore, we invite you to submit a revised version of the manuscript that addresses the points raised during the review process.

**The manuscript focuses on a topic of potential interest. However, the study has several shortcomings that should be addressed. To mention few of them, i) need to provide an explanation on the rational behind using CD68 IHC staining; ii) unclear whether there is any difference in the prevalence of TMA among races; iii) unclear whether there is any difference among races in the progression of kidney failure in those with TMA, iv) unclear whether genetic studies in complement genes were performed to evaluate possible genetic causes of TMA; v) unclear whether patients classified as M0 have other morphologic features indicated in the Oxford Classification; vi) need to provide the clinical course of these M0 patients; vii) need to provide a more detailed description of the morphologic feature of TMA in the Results section; viii) provide detailed information regarding extra-renal target organ damage related to hypertension; ix) need to elaborate on complement dysregulation in patients presenting with hypertensive emergency; x) explore the possibility to stain of C4d and/or MBI to dissect the complement cascade; xi) unclear what means p25 and p75; xii) need to clarify the statement on statistical significance attributed to p-values <0,05; xiii) unclear for the logistic regression what covariates have been used and what the sample size was to train the model.**

We look forward to receiving your revised manuscript.

Kind regards,

Giuseppe Remuzzi

Academic Editor

PLOS ONE

Journal Requirements:

2. Please refer to any post-hoc corrections to correct for multiple comparisons during your statistical analyses. If these were not performed please justify the reasons. Please refer to our statistical reporting guidelines for assistance (https://journals.plos.org/plosone/s/submission-guidelines.#loc-statistical-reporting).

Reviewers' comments:

Reviewer's Responses to Questions

**Comments to the Author**

1. Is the manuscript technically sound, and do the data support the conclusions?

Reviewer #1: Yes

Reviewer #2: Partly

Reviewer #3: Partly

2. Has the statistical analysis been performed appropriately and rigorously? 

Reviewer #1: Yes

Reviewer #2: No

Reviewer #3: Yes

3. Have the authors made all data underlying the findings in their manuscript fully available?

Reviewer #1: Yes

Reviewer #2: No

Reviewer #3: No

4. Is the manuscript presented in an intelligible fashion and written in standard English?

Reviewer #1: Yes

Reviewer #2: Yes

Reviewer #3: Yes

5. Review Comments to the Author

Reviewer #1: The authors describe the prevalence and outcomes of TMA in patients with IgAN in a retrospective single center cohort.

Please consider the following comments and suggestions:

- Please include in Methods the timeline of the data analyzed in this cohort (i.e. This study was conducted in patients from X year through X year)

- Please provide small explanation on the rational behind using CD68 IHC staining for the readers to understand why it was used in these cases.

- Authors mention 73% of subjects were whites. Please mention the different races in the remaining 27% of the cohort population, and add this information to your baseline characteristics table

- Is there any difference in the prevalence of TMA among races?

- Is there any difference among races in the progression of kidney failure in those with TMA?

- Was the cause of TMA investigated in any those 21 patients included for analysis? If so, this should be mentioned in the manuscript

- Did any genetic studies in complement genes were performed to further evaulate possible genetic causes of TMA in these subjects?

Reviewer #2: Variables with non-

146 parametric distributions were expressed as median values (p25 and p75)

What is p25 and p75? Is it the first and third quartile? Please clarify this in detail.

Statistical significance was attributed to p-values <0.05.

I guess the authors mean the significance level was chosen to be alpha=0.05?

https://www.mdpi.com/2504-4990/1/3/54

If so please correct the statement or clarify.

In the results section, the authors present a survival analysis but in the methods section this is not discussed. Please revise this section correspondingly.

See

https://www.mdpi.com/2504-4990/1/3/58

how results should be presented.

For the Logistic regression it is unclear what covariates have been used. Also it is unclear what the sample size was to train the model.

The authors mention that 'These findings indicate that vascular compartment may also be a prognostic marker in IgAN patients.' however it is unclear what is the connection of this to the statistical analysis conducted. Please explain if results from the statistical analysis point to this finding and elaborate on the robustness of this finding.

Reviewer #3: Neves et al. analyzed the prognostic role of TMA on kidney biopsy in 118 patients with IgA nephropathy, a prevalent form of GN with variable prognosis, and state that TMA lesions affect the prognosis and, in particular, the risk of ESKD. However, I do have some points that should be discussed.

- Mesangial IgA deposits on immunofluorescence microscopy are common in the general population and are not always related to disease. This is particular the case in patients with no mesangial proliferation. Neves et al. showed that 76.3% patients had significant mesangial proliferation, that is, M1, indicating that mesangial abnormalities were lacking in about 25% patients. Do these patients classified as M0 have other morphologic features indicated in the Oxford classification, mesangial C3c deposits, and/or electron dense deposits on electron microscopy? Also, the clinical course of these “M0 patients” would be interesting to know if none of the aforementioned morphologic features are present on kidney biopsy to exclude patients with “normal” IgA deposits on immunofluorescence microscopy as such patients may bias the results.

- The authors should provide a more detailed description of the morphologic features of TMA in the Results section. How many patients present with acute and/or chronic lesions? In my experience, patients with chronic features of TMA often have persistent phospholipid serum reactivity, either fulfilling clinical criteria for the antiphospholipid syndrome or not, drug-induced TMA (e.g., anti-VEGF treatment, chemotherapy), or genetic variants linked to low CD46 activity (this has also been found in patients carrying C3 p.R161W), etcetera. It is important to note that patients with a MPGN pattern on light microscopy should not have electron dense deposits along the glomerular capillary wall on electron microscopy.

- Do patients with TMA on kidney biopsy often have complement deposits co-localized with thrombi and/or other lesions related to the TMA?

- Patients with TMA on kidney biopsy invariably presented with hypertension. The TMA was often localized to the kidneys without profound hematologic abnormalities on peripheral blood. This has been observed in patients with TMA presenting with hypertensive emergency and thus, extrarenal target organ damage, e.g., hypertensive retinopathy, left ventricular hypertrophy. Many of such patients may present with complement-mediated TMA. I would suggest to add detailed information regarding extrarenal target organ damage related to hypertension. The authors should elaborate on complement dysregulation in patients presenting with hypertensive emergency. (For example, genotyping if feasible, recurrent disease prior to and after kidney transplantation.)

- Previous studies suggested that activation of the lectin pathway of complement may affect the prognosis. C4d can be found in about 25% patients with IgA nephropathy and appear as prevalent as morphologic features of TMA. MASP2, downstream of MBL, has been implicated in thrombosis. Is it possible to stain of C4d and/or MBL to better dissect the complement cascade?

- Table 1 should be updated. ACE inhibitor or ARB alone indicates no concomitant medication, although >50% patients had been treated with immunosuppressive drugs.

- Table 5 should also include the HRs of hypertension, serum creatinine, eGFR, and the complete Oxford classification (unless the authors only corrected for IF/TA). I would suggest to pick either serum creatinine or eGFR as a confounder instead of both.

6. PLOS authors have the option to publish the peer review history of their article (what does this mean?). If published, this will include your full peer review and any attached files.

Reviewer #1: Yes: MARIA L. GONZALEZ SUAREZ

Reviewer #2: No

Reviewer #3: Yes: Sjoerd A.M.E.G. Timmermans

---

## [Author Response · Author response to Decision Letter 0]

17 Jul 2020

July 16th, 2020

To Dr. Joerg Heber

Editor-in-Chief of PLos One

Dear Editor,

Please find attached the revised version of the manuscript “The impact of histologic evidence of thrombotic microangiopathy in the renal outcomes of patients with IgA nephropathy”, by Neves et al, which is being submitted for the “Original Research Article” section. 

We thank you for the follow-up and the reviewers for the helpful comments and suggestions, which certainly improved our manuscript quality.

We believe that we have appropriately addressed all points raised by the reviewers and that the manuscript is now suitable for publication. The specific responses to the editors and reviewers are outlined below.

REVIEWER 1 REPORT: 

The authors describe the prevalence and outcomes of TMA in patients with IgAN in a retrospective single center cohort. Please consider the following comments and suggestions:

1. Please include in Methods the timeline of the data analyzed in this cohort (i.e. This study was conducted in patients from X year through X year)

Response: Thank you for your point. We have added the information as requested (Page 4, line 91).

2. Please provide small explanation on the rational behind using CD68 IHC staining for the readers to understand why it was used in these cases.

Response: Thank you for your relevant comment. Recent studies (Soares et al, Histopathology. 2019;74(4):629-637; and Trimarchi H et al Kidney Int. 2019;95(4):750-756) have highlighted the relevance of glomerular staining of CD68 as an additional tool to define endocapillary hypercellularity (E) more accurately. (Page 13, lines 333-339).

3. Authors mention 73% of subjects were whites. Please mention the different races in the remaining 27% of the cohort population, and add this information to your baseline characteristics table.

Response: Thank you for your comment. We have described the detailed races in the text (Page 7, line 166-167) and in the Table 1.

4. Is there any difference in the prevalence of TMA among races? 

Response: Thank you for your comment. There was no statistical significant difference in the prevalence of TMA among races, as described in the text (Page 8, line 187) and in the Table 4.

5. Is there any difference among races in the progression of kidney failure in those with TMA?

Response: Thank you for your comment. In patients with histological evidences of TMA, 15 (71.4%) undergone Chronic Kidney Disease requiring dialysis. When we describe dialysis patients according race, 1/7 (14.2%) was east-asian, 4/25 (16%) were black and 10/86 (11.6%) were white. Comparing the proportions in the total population, there were no difference among races in the progression of TMA (p=0.792). (Page 8, line 187)

6. Was the cause of TMA investigated in any those 21 patients included for analysis? If so, this should be mentioned in the manuscript.

Response: Thank you for comment. Among our patients there are no solid organs transplanted patients, bone marrow transplant, pregnant women or patients diagnosed with cancer. We have no description of possible medications associated with TMA. Serologies were performed for Hepatitis B, C and HIV as well as research for autoimmune diseases (ANA, ANCA, Rheumatoid Factor, lupus anticoagulant). No genetic tests have been performed. We have described all these information in the manuscript (Page 4, Lines 95-99).

7. Did any genetic studies in complement genes were performed to further evaluate possible genetic causes of TMA in these subjects?

Response: Thank you for your comment. We are starting in our institution a study of genetic analysis of patients with TMA associated to primary and secondary glomerulopathies. At the moment, patients were not tested but we plan to perform analysis by whole exome sequencing. (Page 14, lines 344-347).

REVIEWER 2 REPORT: 

1. Variables with non- parametric distributions were expressed as median values (p25 and p75). What is p25 and p75? Is it the first and third quartile? Please clarify this in detail.

Response: Thank you for your comment. The terms p25 and p75 refers to first and third quartile. We have corrected it in the Material and Methods section as suggested (Page 6, line 156).

2. Statistical significance was attributed to p-values <0.05. I guess the authors mean the significance level was chosen to be alpha=0.05? https://www.mdpi.com==/2504-4990/1/3/54. If so please correct the statement or clarify.

Response: Thank you for your comment. We have made the correction in the Material and Methods section as requested (Page 7, line 160-161).

3. In the results section, the authors present a survival analysis but in the methods section this is not discussed. Please revise this section correspondingly. See: https://www.mdpi.com/2504-4990/1/3/58

how results should be presented.

Response: Thank you for your raised point. We have described the analysis in the Material and Methods section as suggested (Page 7, lines 161-162).

4. For the Logistic regression it is unclear what covariates have been used. Also it is unclear what the sample size was to train the model.

Response: Thank you for your valuable comment. For Logistic regression we have analyzed data from all 118 patients. We have detailed the covariants in in the Material and Methods section (Page 7, lines 157-160) and in the Table 6 as suggested.

5. The authors mention that 'These findings indicate that vascular compartment may also be a prognostic marker in IgAN patients.' however it is unclear what is the connection of this to the statistical analysis conducted. Please explain if results from the statistical analysis point to this finding and elaborate on the robustness of this finding. 

Response: Thank you for your high relevant comment. In our manuscript we demonstrated that evidences of TMA in renal biopsy were an independent risk factor for reaching CKD-KRT. This finding corroborates previous published data; however, our sample is of 118 patients. In this context, we opted to replace the sentence “These findings indicate that vascular compartment may also be a prognostic marker in IgAN patients” to “Further studies are necessary to analyze the impact of vascular lesions on IgAN prognosis.” (Page 2, Line 49) and (Page 14, lines 356-357).

REVIEWER 3 REPORT: 

Neves et al. analyzed the prognostic role of TMA on kidney biopsy in 118 patients with IgA nephropathy, a prevalent form of GN with variable prognosis, and state that TMA lesions affect the prognosis and, in particular, the risk of ESKD. However, I do have some points that should be discussed.

1. Mesangial IgA deposits on immunofluorescence microscopy are common in the general population and are not always related to disease. This is particular the case in patients with no mesangial proliferation. Neves et al. showed that 76.3% patients had significant mesangial proliferation, that is, M1, indicating that mesangial abnormalities were lacking in about 25% patients. Do these patients classified as M0 have other morphologic features indicated in the Oxford classification, mesangial C3c deposits, and/or electron dense deposits on electron microscopy? Also, the clinical course of these “M0 patients” would be interesting to know if none of the aforementioned morphologic features are present on kidney biopsy to exclude patients with “normal” IgA deposits on immunofluorescence microscopy as such patients may bias the results.

Response: Thank you for your interesting comment. In our sample we have 24 patients (20.3%) that does not fulfill M1 criteria of Oxford Classification. We have made a table to make the analysis of clinical and histological M0 patients easier. The table with clinical, laboratory tests, renal biopsy findings and outcomes of patients with M0 is following:

Table: Clinical profile, laboratory tests, renal biopsy findings and outcomes of patients with M0

Clinical and Laboratory data at renal biopsy Renal Biopsy Findings End of follow-up

Gender SCr (mg/dL) CKD-EPI

(ml/min/1,73m2) Proteinuria

(g/dia) HMT HTN M E S T C TMA IgG IgA C3 IgM C1q SCr 

(mg/dL) CKD-EPI

(ml/min/1,73m2) Dialysis

Male 1,46 52,95 0,81 Yes Yes 0 1 1 0 1 No neg ++ ++ neg neg 0,8 111 No

Male 3,5 17 2,4 Yes Yes 0 0 1 1 1 No + ++ neg neg neg 1,2 58 No

Female 0,8 111 1,5 Yes No 0 0 0 0 0 No neg + neg neg neg 0,65 135 No

Male 0,92 100 0,82 Yes No 0 0 0 0 0 No neg + + + neg 0,72 99,93 No

Female 1,89 26 2,98 Yes Yes 0 0 1 0 1 No neg +++ ++ neg neg 1,07 49,34 No

Female 3,05 18 1,36 Yes Yes 0 0 1 1 0 No neg +++ neg neg neg 3,91 14,27 Yes

Male 1 117 1,27 No No 0 0 0 0 0 No neg + neg + neg 1,17 81,53 No

Female 0,7 92 0,51 Yes No 0 0 0 0 0 No neg + + neg neg 0,62 102 No

Male 1,6 62 9,7 Yes Yes 0 0 1 1 1 No neg ++ + neg neg 5,7 13 Yes

Male 0,75 113,6 1,39 No Yes 0 0 0 0 0 No neg + + neg neg 0,92 94,71 No

Male 1 94 2,35 Yes No 0 0 1 0 0 No neg ++ + neg neg 0,73 120,59 No

Female 0,8 91 0,33 Yes No 0 0 0 0 0 No neg ++ + neg neg 0,6 95 No

Female 0,74 120 2,68 Yes Yes 0 1 0 0 1 No + +++ + neg + 0,81 102,7 No

Male 1,23 81,24 0,09 Yes No 0 0 1 0 0 No neg ++ + neg neg 0,84 105,24 No

Female 2,13 33 0,71 Yes Yes 0 0 0 1 0 Yes neg ++ + neg neg 2,24 26,05 No

Female 1,44 39,73 1,2 Yes Yes 0 0 0 1 0 No neg +++ neg neg neg 1,24 46,78 No

Female 0,99 59,56 2,55 Yes Yes 0 0 1 0 0 No neg ++ neg neg neg 0,7 82 No

Female 0,9 74 8,1 No No 0 0 0 0 0 No neg ++ neg neg neg 0,71 97,06 No

Female 1,2 52 2,5 Yes No 0 0 0 0 0 No neg + + + neg 1,1 55 No

Male 2,6 36 3,3 No Yes 0 0 1 0 0 No neg +++ ++ neg neg 6,96 9 Yes

Male 1,24 71,62 4,7 Yes Yes 0 0 1 1 0 No neg ++ ++ neg neg 0,82 122,88 No

Male 1,2 83 0,5 Yes Yes 0 0 1 0 0 No neg +++ + + neg 0,99 100 No

Male 1,06 86 2,03 Yes No 0 0 1 0 1 No neg ++ + neg neg 12,7 4,91 Yes

Male 1,2 83 0,5 Yes Yes 0 0 1 0 0 No neg +++ ++ + neg 0,99 100 No

HMT: hematúria; HTN: hypertension; Neg: negative; SCr: sérum creatinine; TMA: Thrombotic Microangiopathy

The table shows that all patients with M0 has clinical, laboratory and/or histological evidences of current IgAN disease, what exclude the possibility of normal IgA deposits on renal biopsy.

Normal IgA deposits on immunofluorescence microscopy of patients are scarcely mentioned in literature. Waldherr et al (Nephrol. Dial Transpl. 1989;4(11):943-946), studying 250 consecutive autopsy cases with no renal diagnosed disease showed mesangial deposition in 12 (4.8). From those ones, 6 patients had cirrhosis (a well-known cause of secondary IgAN) while 6 suffered from various other con-ditions including endocarditis, bronchial asthma, cardio-vascular disease, and neoplasia. Two of these patients had completely negative urine analysis on repeated investigations, whereas three patients exhibited microscopic haematuria and/or mild proteinuria. C3c deposits in glomeruli were detected in one kidney.

2. The authors should provide a more detailed description of the morphologic features of TMA in the Results section. How many patients present with acute and/or chronic lesions? In my experience, patients with chronic features of TMA often have persistent phospholipid serum reactivity, either fulfilling clinical criteria for the antiphospholipid syndrome or not, drug-induced TMA (e.g., anti-VEGF treatment, chemotherapy), or genetic variants linked to low CD46 activity (this has also been found in patients carrying C3 p.R161W), etcetera. It is important to note that patients with a MPGN pattern on light microscopy should not have electron dense deposits along the glomerular capillary wall on electron microscopy.

Response: Thank you for your highly relevant comment. We decided to create a new table (Table 2) where we describe the frequencies of histological findings of TMA in renal biopsies of patients with IgAN. We also describe those finding in the manuscript (Page 7, lines 173-180).

3. Do patients with TMA on kidney biopsy often have complement deposits co-localized with thrombi and/or other lesions related to the TMA?

Response: Thank you for your comment. Our pathologist as a routine analysis describes complement deposits only in glomerular compartment, so, there is no information about co-localization of protein complement in TMA lesion. However, in 21 patients described with acute and/or chronic glomerular lesions of TMA, glomerular C3 deposition was evidenced in 18 patients. Most renal biopsy has no available fragment, hence, we will not able to reassess this information.

4. Patients with TMA on kidney biopsy invariably presented with hypertension. The TMA was often localized to the kidneys without profound hematologic abnormalities on peripheral blood. This has been observed in patients with TMA presenting with hypertensive emergency and thus, extrarenal target organ damage, e.g., hypertensive retinopathy, left ventricular hypertrophy. Many of such patients may present with complement-mediated TMA. I would suggest to add detailed information regarding extrarenal target organ damage related to hypertension. The authors should elaborate on complement dysregulation in patients presenting with hypertensive emergency. (For example, genotyping if feasible, recurrent disease prior to and after kidney transplantation.)

Response: Thank you for your comment. Unfortunately we have no data about extrarenal organ damage of most patients. We added to the discussion a paragraph describing about complement dysregulation in patients presenting with hypertensive emergency (Page 11, lines 268-285).

5. Previous studies suggested that activation of the lectin pathway of complement may affect the prognosis. C4d can be found in about 25% patients with IgA nephropathy and appear as prevalent as morphologic features of TMA. MASP2, downstream of MBL, has been implicated in thrombosis. Is it possible to stain of C4d and/or MBL to better dissect the complement cascade?

Response: Thank you for your valuable comment. From 118 patients, 72 had renal biopsy fragments that allowed immunohistochemistry for C4d, being 9/21 with TMA and 63/97 without TMA. The results of Immunohistochemistry for C4d is described in the text (Page 9, lines 216-220), and Table 4.

6. Table 1 should be updated. ACE inhibitor or ARB alone indicates no concomitant medication, although >50% patients had been treated with immunosuppressive drugs.

Response: Thank you for your comment. We have corrected it on Table 4.

7. Table 5 should also include the HRs of hypertension, serum creatinine, eGFR, and the complete Oxford classification (unless the authors only corrected for IF/TA). I would suggest to pick either serum creatinine or eGFR as a confounder instead of both.

Response: Thank you for your comment. We have detailed the covariants analyzed in in the Material and Methods section (Page 7, lines 157-160) and in Table 6 as suggested.

Sincerely,

Precil Diego Miranda de Menezes Neves, MD

Division of Nephrology

University of São Paulo School of Medicine

São Paulo - Brazil

---

## [Decision Letter · Decision Letter 1]

27 Aug 2020

PONE-D-20-12431R1

Evidences of histologic Thrombotic Microangiopathy and the impact in renal outcomes of patients with IgA nephropathy

PLOS ONE

Dear Dr. Neves,

Thank you for submitting your manuscript to PLOS ONE. After careful consideration, we feel that it has merit but does not fully meet PLOS ONE’s publication criteria as it currently stands. Therefore, we invite you to submit a revised version of the manuscript that addresses the points raised during the review process.

**The revised version of the manuscript is improved. However, few additional issues remain to be addressed. In particular, i) recommendation to exclude patients with no mesangial hypercellularity (M0) and no other morphologic features included in the MEST-C classification; ii) change the results section (page 9, line 218) as indicated by Reviewer 3.**

We look forward to receiving your revised manuscript.

Kind regards,

Giuseppe Remuzzi

Academic Editor

PLOS ONE

Reviewers' comments:

Reviewer's Responses to Questions

**Comments to the Author**

1. If the authors have adequately addressed your comments raised in a previous round of review and you feel that this manuscript is now acceptable for publication, you may indicate that here to bypass the “Comments to the Author” section, enter your conflict of interest statement in the “Confidential to Editor” section, and submit your "Accept" recommendation.

Reviewer #1: All comments have been addressed

Reviewer #3: (No Response)

2. Is the manuscript technically sound, and do the data support the conclusions?

Reviewer #1: Yes

Reviewer #3: Yes

3. Has the statistical analysis been performed appropriately and rigorously? 

Reviewer #1: I Don't Know

Reviewer #3: Yes

4. Have the authors made all data underlying the findings in their manuscript fully available?

Reviewer #1: Yes

Reviewer #3: Yes

5. Is the manuscript presented in an intelligible fashion and written in standard English?

Reviewer #1: No

Reviewer #3: Yes

6. Review Comments to the Author

Reviewer #1: Thank you for addressing reviewers comments. It is an interesting topic, and I think this revised version has improved manuscript content.

Reviewer #3: Almost all comments have been addressed appropriately. However, I would recommend to exclude patients with no mesangial hypercellularity (M0) and no other morphologic features included in the MEST-C classification. This is particular the case if electron dense deposits were not found/electron microscopy was not performed.

Also, I would sugges to change the results section page 9 line 218 into: In total, 72 kidney biopsies were available to stain for C4d, including 9 tissue sections from patients with TMA and 63 tissue sections from those without TMA. Glomerular deposits of C4d were more frequent in patients with TMA as compared to those without TMA (n/M=7/9 and n/N=26/63; P=0.04).

7. PLOS authors have the option to publish the peer review history of their article (what does this mean?). If published, this will include your full peer review and any attached files.

Reviewer #1: **Yes: **Maria L. Gonzalez Suarez

Reviewer #3: **Yes: **Sjoerd A.M.E.G. Timmermans

---

## [Author Response · Author response to Decision Letter 1]

18 Sep 2020

September 16th, 2020

To Dr. Joerg Heber

Editor-in-Chief of PLos One

Dear Editor,

Please find attached the revised version of the manuscript “The impact of histologic evidence of thrombotic microangiopathy in the renal outcomes of patients with IgA nephropathy”, by Neves et al, which is being submitted for the “Original Research Article” section. 

We thank you for the follow-up and the reviewers for the helpful comments and suggestions, which certainly improved our manuscript quality.

We believe that we have appropriately addressed all points raised by the reviewers and that the manuscript is now suitable for publication. The specific responses to the editors and reviewers are outlined below.

REVIEWER 1 REPORT: 

Thank you for addressing reviewers comments. It is an interesting topic, and I think this revised version has improved manuscript content.

Response: Thank you for your kind comment. Improve the quality of our manuscript was just possible once you have been contributed with your valuable suggestions.

REVIEWER 3 REPORT: 

1. Almost all comments have been addressed appropriately. However, I would recommend to exclude patients with no mesangial hypercellularity (M0) and no other morphologic features included in the MEST-C classification. This is particular the case if electron dense deposits were not found/electron microscopy was not performed.

Response: Thank you for your relevant comment We do understand your concern including patients with MEST-C score zero (M0/E0/S0/T0/C0) in our protocol particularly lacking electronic microscopy analysis. Those patients could be misdiagnosed as IgAN instead of C3 Glomerulopathy (C3GP) . It is well defined that light microscopy (LM) and immunofluorescence analysis (IF) are sufficient for IgAN diagnosis while MEST-C score adds morphological parameters to clinical prognosis and not to diagnosis.

MEST-C score shows high variation among studies. In Valiga study (N=647) and North American Validation study (N=87) M0 is described in 73% to 10% of the patients, respectively, E0 from 88% to 69%, S0 from 25% to 35%, T0 from 78% to 80%. C parameter was added to MEST score recently and C0 is described in 80 to 90 % of the patients. 

In our 118 IgAN patients, MEST-C parameters distribution M0 20.4% , E0 64.4% , S0 29.7% , T0 61.7% , C0 71.2 % was not different from world’s report while 8 patients showed MEST-C score zero in each parameter concurrently. Three out of those 8 patients did not show any C3 deposition making impossible the hypothetical diagnosis of C3GP that is defined as “C3 dominant and at least two orders of intensity stronger than any combination of IgG, IgM, IgA and C1q “ ( Jean Hou et al).

So far, 5 patients with C3 deposition and MEST-C score concurrently zero could be on hazard of a misdiagnosis of IgAN instead of C3GP (see table below describing each patient)

Going deeply on G3GP clinical features described on literature we can emphasize some characteristics :

• C3GP is an uncommon glomerulopathy when compared to IgAN.

• C3GP more frequent light microscopy pattern is Membranoproliferative (MPGN ) from 73% to 78 % (Caravaca-Fontan et al, and Avasare et al) while IgAN is mostly Mesangioproliferative, and in some patients depicts a FSGS pattern. No IgAN patient from our protocol discloses MPGN pattern.

• Considering IF findings in C3GP Avasare et al showed exclusive C3 deposition in 33%, C3+IgM in 17% and do not mention important IgA deposition while in IgAN that deposition is mandatory as disclosed in all our patients.

• Clinical evolution and prognosis are very different between IgAN and C3GP. Our questioned 5 patients showed on last follow-up visit , around 4 years, almost normal kidney function as expected in many non-progressive IgAN patients much different from C3GP patients reaching aroud 50% of kidney failure on the same time 

REFERENCES 

1. Jean Hou et al. Toward a working definition of C3 glomerulopathy by immunofluorescence. KI 85: 450-456, 2014

2. Caravaca-Fontan et al Mycophenolate Mofetil in C3 glomerulopathy and pathogenic drivers of the disease CJASN 15: 1287-1298, 2020

3. Avasare et al Mycophenolate Mofetil in combination with steroids for treatment ok C3 glomerulopathy : a case series CJASN 13: 406-413, 2018

We hope having cleared your question. Nevertheless we would be glad processing EM on those specific 5 patients. Unfortunately we have no biopsy material left.

We are Very gratefull for your comments that highly enriched our paper.

Table: Clinical profile, laboratory tests, renal biopsy findings and outcomes of patients with M0, E0, S0, T0, C0 and C3+

Clinical and Laboratory data at renal biopsy Renal Biopsy Findings End of follow-up

N Gender SCr (mg/dL) CKD-EPI

(ml/min/1,73m2) Proteinuria

(g/dia) HMT HTN M E S T C TMA IgG IgA C3 IgM C1q SCr 

(mg/dL) CKD-EPI

(ml/min/1,73m2) Dialysis

4 Male 0,92 100 0,82 Yes No 0 0 0 0 0 No neg + + + neg 0,72 99,93 No

8 Female 0,7 92 0,51 Yes No 0 0 0 0 0 No neg + + neg neg 0,62 102 No

10 Male 0,75 113,6 1,39 No Yes 0 0 0 0 0 No neg + + neg neg 0,92 94,71 No

12 Female 0,8 91 0,33 Yes No 0 0 0 0 0 No neg ++ + neg neg 0,6 95 No

19 Female 1,2 52 2,5 Yes No 0 0 0 0 0 No neg + + + neg 1,1 55 No

HMT: hematúria; HTN: hypertension; Neg: negative; SCr: sérum creatinine; TMA: Thrombotic Microangiopathy

2. Also, I would suggest to change the results section page 9 line 218 into: In total, 72 kidney biopsies were available to stain for C4d, including 9 tissue sections from patients with TMA and 63 tissue sections from those without TMA. Glomerular deposits of C4d were more frequent in patients with TMA as compared to those without TMA (n/M=7/9 and n/N=26/63; P=0.04).

Response: Thank you for your suggestion. We have replaced the sentence as suggested (Page 9, lines 216-220).

Sincerely,

Precil Diego Miranda de Menezes Neves, MD

Division of Nephrology

University of São Paulo School of Medicine

São Paulo - Brazil

---

## [Decision Letter · Decision Letter 2]

1 Oct 2020

Evidences of histologic Thrombotic Microangiopathy and the impact in renal outcomes of patients with IgA nephropathy

PONE-D-20-12431R2

Dear Dr. Neves,

We’re pleased to inform you that your manuscript has been judged scientifically suitable for publication and will be formally accepted for publication once it meets all outstanding technical requirements.

**The re-revised manuscript is definitely improved. The authors have adequately addressed all the remaining comments**

Kind regards,

Giuseppe Remuzzi

Academic Editor

PLOS ONE

Additional Editor Comments (optional):

Reviewers' comments:

Reviewer's Responses to Questions

**Comments to the Author**

1. If the authors have adequately addressed your comments raised in a previous round of review and you feel that this manuscript is now acceptable for publication, you may indicate that here to bypass the “Comments to the Author” section, enter your conflict of interest statement in the “Confidential to Editor” section, and submit your "Accept" recommendation.

Reviewer #1: All comments have been addressed

Reviewer #3: All comments have been addressed

2. Is the manuscript technically sound, and do the data support the conclusions?

Reviewer #1: Yes

Reviewer #3: Yes

3. Has the statistical analysis been performed appropriately and rigorously? 

Reviewer #1: I Don't Know

Reviewer #3: Yes

4. Have the authors made all data underlying the findings in their manuscript fully available?

Reviewer #1: Yes

Reviewer #3: Yes

5. Is the manuscript presented in an intelligible fashion and written in standard English?

Reviewer #1: Yes

Reviewer #3: Yes

6. Review Comments to the Author

Reviewer #1: Thank you for submitting this manuscript after addressing comments. All my comments and suggestions have been addressed.

Reviewer #3: (No Response)

7. PLOS authors have the option to publish the peer review history of their article (what does this mean?). If published, this will include your full peer review and any attached files.

Reviewer #1: **Yes: **MARIA L. GONZALEZ SUAREZ

Reviewer #3: **Yes: **Dr. Sjoerd A.M.E.G. Timmermans

---

## [Editor Report · Acceptance letter]

21 Oct 2020

PONE-D-20-12431R2 

Evidences of histologic Thrombotic Microangiopathy and the impact in renal outcomes of patients with IgA nephropathy 

Dear Dr. Neves:

I'm pleased to inform you that your manuscript has been deemed suitable for publication in PLOS ONE. Congratulations! Your manuscript is now with our production department. 

Kind regards, 

on behalf of

Prof. Giuseppe Remuzzi 

Academic Editor

PLOS ONE